# Decreased CTRP3 Plasma Concentrations Are Associated with Sepsis and Predict Mortality in Critically Ill Patients

**DOI:** 10.3390/diagnostics9020063

**Published:** 2019-06-21

**Authors:** Eray Yagmur, Simone Otto, Ger H. Koek, Ralf Weiskirchen, Christian Trautwein, Alexander Koch, Frank Tacke

**Affiliations:** 1Medical Care Center, Dr. Stein and Colleagues, D-41169 Mönchengladbach, Germany; 2Department of Medicine III, RWTH-University Hospital Aachen, D-52074 Aachen, Germany; simone.otto1@rwth-aachen.de (S.O.); ctrautwein@ukaachen.de (C.T.); akoch@ukaachen.de (A.K.); ftacke@ukaachen.de (F.T.); 3Section of Gastroenterology and Hepatology, Department of Internal Medicine, Maastricht University Medical Medical Centre (MUMC), 6202AZ Maastricht, The Netherlands; gh.koek@mumc.nl; 4Institute of Molecular Pathobiochemistry, Experimental Gene Therapy and Clinical Chemistry, RWTH-University Hospital Aachen, D-52074 Aachen, Germany; rweiskirchen@ukaachen.de; 5Department of Hepatology and Gastroenterology, Charité University Medical Center, D-10117 Berlin, Germany

**Keywords:** C1q/TNF-related protein 3, CTRP3, ICU, critical illness, sepsis, inflammation, glucose metabolism, insulin resistance, adipokine, diabetes

## Abstract

C1q/ tumor necrosis factor (TNF)-like protein 3 (CTRP3) represents a novel member of the adipokine family that exerts favorable metabolic actions in humans. However, the role of CTRP3 in critical illness and sepsis is currently unknown. Upon admission to the medical intensive care unit (ICU), we investigated CTRP3 plasma concentrations in 218 critically ill patients (145 with sepsis, 73 without sepsis). Results were compared with 66 healthy controls. CTRP3 plasma levels were significantly decreased in critically ill patients, when compared to healthy controls. In particular, low CTRP3 levels were highly associated with the presence of sepsis. CTRP3 levels were neither associated with obesity nor diabetes. In critically ill patients, CTRP3 plasma concentrations were inversely correlated with inflammatory cytokines and classical sepsis markers. Among a wide group of adipokines, CTRP3 only correlated with circulating resistin. Low CTRP3 plasma levels were associated with the overall mortality, and CTRP3 levels below 620.6 ng/mL indicated a particularly increased mortality risk in ICU patients. Our study demonstrates for the first time the role of circulating CTRP3 as a biomarker in critically ill patients that might facilitate diagnosis of sepsis as well as prognosis prediction. The association between low CTRP3 and increased inflammation warrants further pathophysiological investigations.

## 1. Introduction

Critical illness is associated with alterations in secretory and metabolic functions of adipose tissue [1,2,3,4]. In principle, the endocrine productions of the adipose tissue (i.e., adipocytokines or adipokines), are involved in a wide range of processes including dietary intake and appetite regulation, energy expenditure, insulin resistance, lipid metabolism, immunity, inflammatory and acute phase responses, vascular homeostasis, endothelial function, and angiogenesis [5,6,7,8,9]. Based on this wide range of physiological functions, alterations of adipokines have been convincingly linked to inflammatory responses and metabolic alterations that occur during critical illness [2,3].

To date, more than 50 adipokines have been reported [10]. Adipokines with recognized functions in critical illness are leptin, retinol binding protein 4 (RBP4), adiponectin, ghrelin, resistin, and C1q/tumor necrosis factor (TNF) related protein 1 (CTRP1) [1,11,12,13,14,15,16]. Despite its potential involvement in the pathogenesis of inflammation and metabolism, some of these adipokines demonstrate interesting potential as prognostic biomarkers, as their circulating levels are associated with the short- or long-term survival in critically ill patients [4,11,12,13,14,15,16]. Nevertheless, a number of potential confounders must be considered for adipokines, including the etiology of critical illness (e.g., sepsis), pre-existing diabetes, obesity, and organ failure [4,11,12,13,14,15,16].

Complement C1q/tumor necrosis factor (TNF) related protein 3 (CTRP3) is a novel adipokine mainly expressed in subcutaneous and visceral adipose tissue [17,18,19,20]. Like other adipokines, the CTRP family is involved in a variety of clinical and pathophysiological processes including metabolism, food intake, inflammation, tumor metastasis, apoptosis, vascular disorders, and ischemic injury responses [21,22]. Similarly, CTRP3 is considered to have multiple, primarily “beneficial” effects, including lowering glucose levels, inhibiting gluconeogenesis in the liver, increasing angiogenesis, and anti-inflammation [23,24,25]. Its expression is upregulated by insulin and downregulated by chronic lipopolysaccharide exposure [26]. Circulatory CTRP3 levels are reduced in human and animal models of obesity and diabetes [2,17,21].

In this work, we investigated the clinical and prognostic relevance of CTRP3 plasma concentrations in a large cohort of critically ill patients (septic and non-septic patients) from a medical intensive care unit (ICU).

## 2. Materials and Methods

The study included 218 critically ill patients, who were admitted to the medical ICU at the RWTH University Hospital Aachen, Germany. The current cohort of patients was collected from an ongoing, prospective observational trial in our unit, in which patients were included consecutively. For current analysis, we therefore randomly enrolled 218 (*n* = 218) patients that had been treated between 2006 and 2011 from the existing biobank. We excluded patients, who had an elective procedure or were admitted for post-interventional observational stay [27]. We used the Third International Consensus Definitions for Sepsis and Septic Shock as a post-hoc definition for sepsis patients; the others were categorized as non-sepsis patients [28]. The patients were treated following the current guidelines for treatment of sepsis (Surviving Sepsis Campaign) [29]. As a healthy control group, we analyzed 66 (*n* = 66) blood donors (43 male, 23 female, median age 29.5 years, range 18–67 years, body mass index (BMI) median 25.4 kg/m^2^, range 17.9–37 kg/m^2^). None of the investigated controls had apparent disturbed blood counts or elevated values of liver enzymes. All healthy controls had a negative serology for viral hepatitis and human immunodeficiency virus (HIV) [30]. Obesity was classified in agreement with the recommended World Health Organization (WHO) BMI value ≥30 kg/m^2^ (obesity class I, BMI 30.0–34.99 kg/m^2^) that is primarily based on the association between BMI and mortality [31].

To assess the patients’ long-term outcome, we contacted the patients, their relatives and/or the primary care physician in approximately six-month intervals after discharge from the hospital for two years [30].

The study protocol was approved by the local ethics committee and conducted in accordance to the ethical standards laid down in the Declaration of Helsinki (ethics committee of the University Hospital Aachen, RWTH-University Aachen, Germany, reference number EK 150/06). All included participants provided written informed consent.

Blood samples were collected directly at the time of admission to the ICU (before specific therapeutic measures). After centrifugation at 4 °C for 10 min, serum and plasma aliquots of 1 mL were frozen immediately at −80 °C. Plasma CTRP3 concentrations were determined using a quantitative sandwich enzyme immunoassay (ELISA), according to the manufacturer’s instructions (CTRP3 (human) competitive ELISA Kit, #AG-45A-0042TP-KI01, AdipoGen, Liestal, Switzerland). Measurements of the other adipocytokines and related proteins CTRP1, leptin, RBP4, adiponectin, ghrelin, and resistin were included, as previously reported [11,12,13,14,15,16].

Data are given as median and range due to the skewed distribution of most of the parameters. Box plot graphics are used to illustrate differences between subgroups. Since most samples were not normally distributed, the Mann–Whitney U-test was applied to test for statistically significant differences between the two groups. Correlations were assessed by the Spearman’s rank correlation method. All values, including outside values as well as far-out values, were included. *P*-values less than 0.05 were considered as statistically significant. The prognostic value of CTRP3 on the outcome was evaluated by Cox regression models and survival curves were generated by Kaplan–Meier analyses with a CTRP3 cut-off level calculated via the Youden index [32]. All analyses were conducted using IBM SPSS Statistics (SPSS; Chicago, IL, USA).

## 3. Results

### 3.1. CTRP3 Plasma Levels Are Significantly Decreased in Critically Ill Patients as Compared with Healthy Controls

In critical illness, many adipokines are significantly elevated in the blood circulation, such as ghrelin, resistin, and CTRP1 blood concentrations [14,15,16]. On the contrary, we found that CTRP3 plasma levels were significantly decreased in a large cohort of 218 critically ill patients (median 545.1 ng/mL, range 82.9–2395.3 ng/mL; Table 1) at admission to the ICU as compared with 66 healthy controls (median 1088,4 ng/mL, range 539.2–2547.9 ng/mL, *p* < 0.001; Figure 1a).

### 3.2. Reduced CTRP3 Plasma Levels in Critically Ill Patients Are Associated with the Presence of Sepsis

CTRP3 specifically blocks the binding of lipopolysaccharide (LPS) to its receptor, the toll-like receptor 4 (TLR4) [33,34], thereby inhibiting inflammatory responses in innate immune cells [33]. Within the cohort of ICU patients, plasma concentrations of CTRP3 were significantly decreased in patients with sepsis (*n* = 145, median 493.4 ng/mL, range 82.9–2395.3 ng/mL) as compared to patients without sepsis (*n* = 73, median 758.8 ng/mL, range 260.4–2269.1 ng/mL, *p* < 0.001; Figure 1a). Typical sites of infection in sepsis are pneumonia and abdominal and urogenital tract infections, while non-sepsis causes of critical illness include, among others, cardiopulmonary diseases, acute pancreatitis, and decompensated liver cirrhosis (Table 2). Among the septic or non-septic critically ill patients, there was no association between CTRP3 plasma concentrations and these different disease etiologies leading to ICU admission (data not shown).

### 3.3. CTRP3 Plasma Levels in Critically Ill Patients Are Not Associated with Diabetes and Obesity

CTRP3 is primarily involved in glucose metabolism, supporting that its plasma levels are associated with type 2 diabetes and obesity [17]. We therefore assessed whether metabolic comorbidities, including pre-existing obesity or diabetes, impacted CTRP3 levels at ICU admission. Surprisingly, neither pre-existing type 2 diabetes nor obesity, as defined by a body mass index (BMI) above 30 kg/m^2^, were associated with CTRP3 plasma concentrations (Figure 1b,c). Moreover, serum glucose at ICU admission (*r* = −0.126, *p* = 0.064) or glycosylated hemoglobin A1 (HbA1c) (*r* = −0.051, *p* = 0.671) did not correlate with CTRP3 levels in critically ill patients (Table 3). CTRP3 did not show any correlations with other key markers of glucose metabolism such as insulin or the homeostasis model assessment-insulin resistance (HOMA-IR) in ICU patients (data not shown). However, β-cell function (HOMA-β) (*r* = −0.371, *p* = 0.002) and C-peptide (*r* = −0.281, *p* = 0.020) correlated inversely with CTRP3 (Table 3).

### 3.4. CTRP3 Levels in Critically Ill Patients Are Inversely Correlated with Biomarkers of Inflammatory Responses in Critically Ill Patients

Adipose tissue inflammation attributes to dysregulated production and release of inflammatory cytokines and adipokines, including IL-6 and TNF-α as well as leptin, resistin, and adiponectin [35]. We investigated the potential association between CTRP3 and inflammatory responses in critically ill patients. In agreement with the proposed anti-inflammatory function of CTRP3, we observed an inverse correlation between CTRP3 and classical markers of inflammation or sepsis, such as interleukin 6 (*r* = −0.266, *p* = 0.001, Figure 2a), procalcitonin (r = −0.207, *p* = 0.009, Figure 2b), C-reactive protein (*r* = −0.390, *p* < 0.001), TNF-α (*r* = −0.399, *p* = 0.018), and soluble CD87 (soluble urokinase-type plasminogen activator receptor (suPAR), *r* = −0.251, *p* = 0.003) (Table 3). We found no correlation to the anti-inflammatory interleukin 10 (*r* = −0.085, *p* = 0.382). Among a broad range of adipokines, including CTRP1, leptin, RBP4, adiponectin, ghrelin, and resistin [11,12,13,14,15,16], CTRP3 only correlated, inversely, with resistin (*r* = −0.397, *p* = 0.002, Figure 2c).

Circulating CTRP3 displayed no association with markers of renal failure, such as creatinine, cystatin C, and the glomerular filtration rate (GFR), or markers reflecting liver damage and cholestasis like alanine aminotransferase (ALT) and total bilirubin. However, CTRP3 levels correlated with lipid metabolism as reflected by cholesterol, low-density lipoprotein (LDL) cholesterol and high-density lipoprotein (HDL) cholesterols (Table 3).

### 3.5. Low CTRP3 Plasma Levels in Critically Ill Patients Are Associated with Adverse Prognosis

Circulating adipokines have been previously suggested as biomarkers for disease severity as well as short- and long-term survival in various critical illness conditions [2,3,4]. In fact, CTRP3 plasma levels correlated inversely with sequential organ failure assessment (SOFA; *r* = −0.237, *p* = 0.007, Table 3), but not with other ICU scores like acute physiology and chronic health II (APACHE II; *r* = −0.063, *p* = 0.378) or simplified acute physiology score 2 (SAPS2; *r* = −0.187, *p* = 0.129) scores (Table 3).

Critically ill patients have a high risk of mortality, not only during the ICU treatment but also after successful discharge from the ICU [36]. We were able to assess the long-term survival in 207 out of 218 patients by contacting the patients or their relatives during the first three years after ICU discharge, thereby providing a comprehensive picture of overall mortality (during ICU and during follow-up). CTRP3 levels at ICU admission were significantly lower in patients that subsequently died (*n* = 89) compared with survivors (*n* = 118) (Figure 3a). Cox regression analysis revealed that CTRP3 levels at ICU admission were a significant predictor of overall mortality (*p* = 0.020). This finding was corroborated by Kaplan–Meier survival curves analyses, demonstrating that patients with CTRP3 plasma levels of the upper quartile (>75%, corresponding to >848 ng/mL) had the best survival rates, while patients with admission CTRP3 levels of the lower quartile (<25%, corresponding to <387.4 ng/mL) had an unfavorable outcome (Figure 3b). Using the calculated optimal cut-off for CTRP3 of 620.6 ng/mL, patients with low CTRP3 demonstrated a high mortality rate, as depicted by Kaplan–Meier survival curve analysis (Figure 3c).

## 4. Discussion

In this study, we demonstrate that circulating levels of the novel adipokine family member CTRP3 has potential as a biomarker in critically ill patients. Critically ill patients had significantly reduced CTRP3 levels compared with healthy controls, and CTRP3 was particularly low in ICU patients with sepsis. Thus, CTRP3 is the only adipocytokine in septic intensive care patients that is significantly decreased compared to published adipokines and circulating soluble metabolic proteins such as CTRP1, ghrelin, resistin, RBP4, leptin, and ghrelin [11,12,13,14,15,16]. Values below 620.6 ng/mL might indicate a potentially severe progression of critical illness at the ICU. Metabolic comorbidities did not impact CTRP3 in the setting of critical illness. Moreover, low CTRP plasma concentrations predicted the overall mortality in critically ill patients.

Critically ill patients show dramatic metabolic and inflammatory derangements, which include dysregulated adipokines [1,2]. Several observations support that dysregulated adipokines are involved in the pathogenesis of critical illness and sepsis [3,4]. However, adipokine levels might be prone to confounding factors such as age, gender, disease severity, feeding protocols, baseline body fat mass, and nutritional status [2]. However, there is evidence that the presence of a systemic inflammatory response is associated with malnutrition such as increased weight loss, an elevated resting energy expenditure and loss of lean tissue and functional decline. This malnutrition is related with hypoalbuminemia. Thus, in our ICU cohort the close association between albumin and CTRP3 reflects both the loss of the amount of lean tissue and systemic inflammatory response. CTRP3 might potentially be associated with overall nutritional status. Nonetheless, the adipose tissue secretes numerous adipokines that contribute to a wide array of biological and clinical processes [11,12,13,14,15,16]. Thus, adipose tissue might play a major role in metabolic alterations of critical illness. For instance, increased proinflammatory cytokines are expressed in hypertrophied adipocytes and adipose tissue resident immune cells [37].

CTRP3 is particular among adipokines, because it has been associated with manifold beneficial metabolic as well as anti-inflammatory functions. In an animal model of LPS-induced sepsis, CTRP3 overexpression protected against myocardial dysfunction [38,39]. Furthermore, LPS inhibits adipose tissue differentiation, induces insulin resistance, and prevents the expression of CTRP3 [40]. CTRP3 specifically blocks the binding of LPS to its receptor TLR4 [33,34] and inhibits proinflammatory responses [33]. However, neither chronic CTRP3 deficiency nor overexpression altered the inflammatory response to a sublethal challenge to LPS [25]. The anti-inflammatory effect of CTRP3 has been rather characterized by decreased mRNA levels of TNF-α and IL-6 [2,3,4,17,21]. It is known that low-grade inflammation in adipose tissue attributes to dysregulated production and release of cytokines and adipokines, including IL-6, TNF-α, monocyte chemotactic protein (MCP)-1, leptin, resistin, and adiponectin [35]. In adipose tissue, weight cycling leads to significantly decreased CTRP3 mRNA expression and impaired glucose metabolism and insulin sensitivity by decreasing CTRP3 [26]. In our study, circulating CTRP3 correlated inversely with several inflammatory markers and cytokines, supporting a counter-regulatory mechanism in critical illness and sepsis. However, we cannot conclude from these associations whether low CTRP3 levels are an important physiological response to allow inflammation or whether inadequately low CTRP3 contributes to disease pathogenesis in ICU patients. The close association between low CTRP3 and adverse prognosis, however, supports the latter hypothesis. Basic research in experimental sepsis models should investigate the functional contribution of low CTRP3 to excessive inflammation as well as the potential to therapeutically target CTRP3 in this setting.

Another aspect of CTRP3 biology is the promotion of insulin resistance by activating inflammatory signaling pathways of c-Jun N-terminal kinase and inhibitor of kB kinase [41]. This attenuates insulin effectiveness by serine phosphorylation of the insulin receptor substrate protein-1 [42]. Impaired glucose tolerance and insulin sensitivity leads to decreased CTRP3. In an experimental animal model, both CTRP3 overexpression and daily CTRP3 administration were effective in regulating high-fat diet-induced hepatic insulin resistance and hepatic steatosis [43], but not in mice fed a low-fat diet [43,44]. These data suggest that the metabolic effect of CTRP3 is specific to the liver, as no changes to metabolism is observed in skeletal muscle in any experimental model examined. CTRP3 may function specifically to regulate metabolism in response to elevated lipid consumption [26,43,44]. In our study, we found some correlations of CTRP3 with lipid metabolism, beta-cell function, and C-peptide, but no association between CTRP3 and diabetes or obesity. This is in principal unexpected, based on CTRP3’s role in glucose homeostasis [17]. However, CTRP3 levels have been reported to be elevated [45], unaltered [46,47], or reduced [48,49,50] in individuals with obesity and/or diabetes.

While our study established a role of circulating CTRP3 as a biomarker in critical illness, several limitations need to be mentioned. Although our study comprises a heterogeneous cohort of “real-life” medical ICU patients, the single-center design and the limited number of study subjects may not allow to conduct multiple regression and very detailed subgroup analyses. Thus, our findings need to be validated in larger, multi-center analyses. In addition, the close association between CTRP3 and inflammation, as well as prognosis, are at this point descriptive. Nonetheless, our data provide a strong rationale for investigating the function involvement of CTRP3 in systemic inflammation and sepsis. Moreover, we found a broad variety of correlations with inflammation, coagulation and others. This is in line with the currently accepted characteristic of this member of the CTRP superfamily that has been documented to have a wide range of effects on metabolism, food intake, inflammation, vascular disorders. Thus, there are many aspects of CTRP3’s regulation and function that have to be explored [17].

## 5. Conclusions

CTRP3 is a novel member of the adipokine family, linking inflammatory and metabolic diseases. Our comprehensive analysis of CTRP3 plasma concentrations in a large, prospectively enrolled cohort of critically ill medical patients support that CTRP3 is an interesting biomarker in this setting. Low CTRP3 values may have diagnostic implications by pointing towards inflammatory and infectious diseases as well as prognostic implications by indicating a high risk of mortality. Thereby, CTRP3 appears to be integrated in the tightly regulated and complex network of adipose tissue-derived endocrine mediators during critical illness. However, CTRP1 does not yet have a clear clinical benefit, but the recognition of potential correlation between metabolic changes and inflammation in intensive care patients could indicate a clinical relevance of this adipokine. Its functional contribution and validation as a prognostic biomarker warrants further investigation.

## Figures and Tables

**Figure 1 diagnostics-09-00063-f001:**
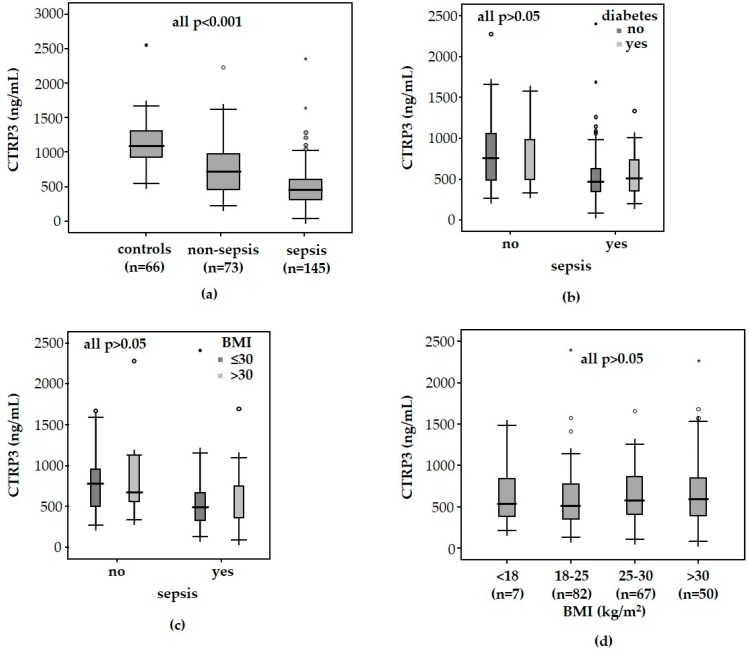
CTRP3 levels in critically ill patients. CTRP3 plasma concentrations are significantly decreased in critically ill patients compared with healthy controls (**a**). ICU patients with sepsis displayed significantly decreased CTRP3 levels compared to patients without sepsis (**a**). CTRP3 plasma concentrations in ICU patients are neither associated with pre-existing type 2 diabetes (**b**) nor obesity, as defined by a body mass index (BMI) above 30 kg/m^2^ (**c**). ICU patients did not show significant differences in reference to BMI classification (**d**). *P*-values (U-test) are given in the figure.

**Figure 2 diagnostics-09-00063-f002:**
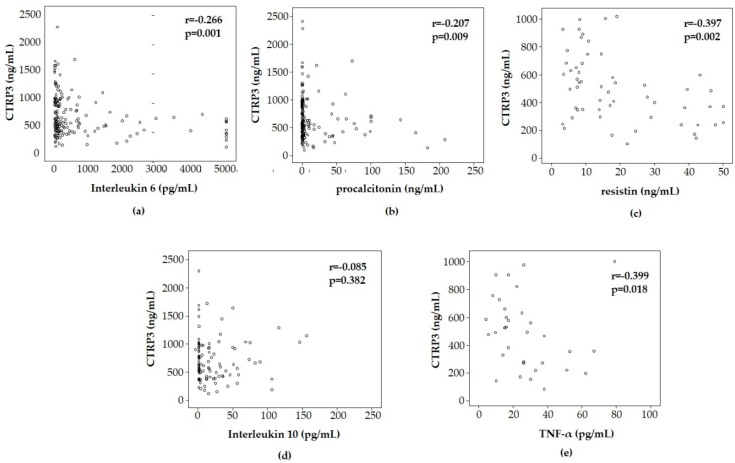
CTRP3 levels correlate inversely with inflammatory response (Spearman rank correlation test). CTRP3 plasma levels, at time of admission to the ICU, are correlated inversely with inflammatory biomarkers such as serum interleukin 6 (**a**), procalcitonin (**b**), and TNF-α (**e**), but not IL-10 (**d**). CTRP3 correlated inversely with serum resistin concentrations in ICU patients (**c**). TNF: tumor necrosis factor; IL-10: interleukin 10.

**Figure 3 diagnostics-09-00063-f003:**
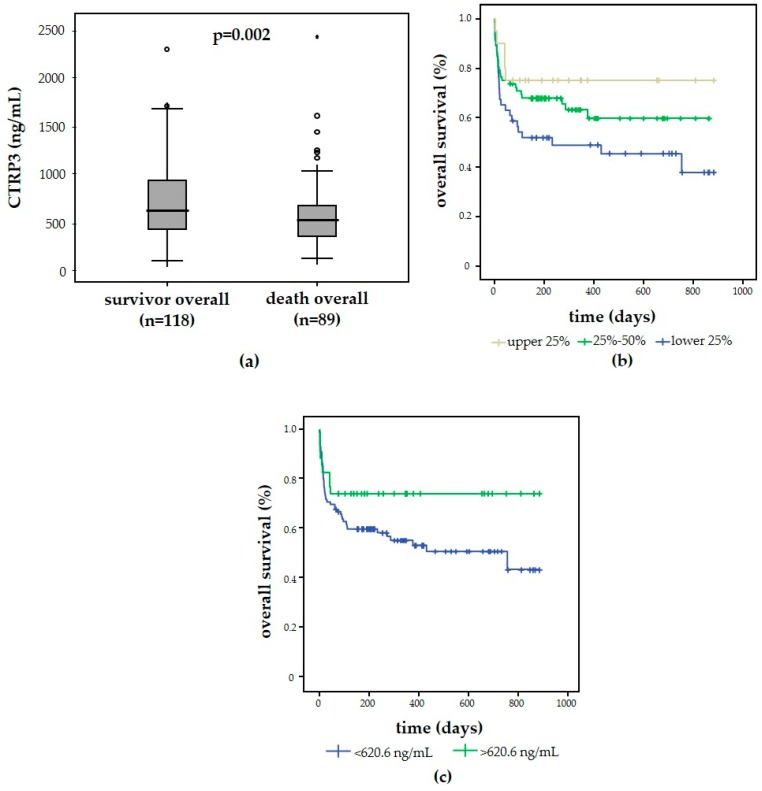
CTRP3 is a biomarker for mortality in critically ill patients. Patients that subsequently died displayed significantly lower CTRP3 levels at ICU admission compared to patients that survived in the long-term follow-up (**a**). High vs. low CTRP3 plasma concentrations discriminated survival of the critically ill patients, as displayed by Kaplan–Meier survival curve analysis for CTRP3 separated into quartiles (**b**). Low CTRP plasma concentrations at ICU admission (optimal cut-off: 620.6 ng/mL) predicted the overall mortality in critically ill patients (**c**).

**Table 1 diagnostics-09-00063-t001:** Baseline patient characteristics and C1q/ tumor necrosis factor (TNF)-like protein 3 (CTRP3) plasma measurements.

Parameter	All Patients	Non–Sepsis	Sepsis	* *p*
Number (*n*)	218	73	145	
Sex (male/female) (*n*)	133/85	48/25	85/60	n.s.
Age (years)	64 (18–90)	61 (18–85)	65 (20–90)	n.s.
APACHE-II score	18 (2–43)	13.5 (2–33)	19 (4–43)	<0.001
SOFA score	9 (0–19)	9.5 (2–19)	7 (0–17)	0.002
Intensive care unit (ICU) days	7 (1–137)	6 (1–45)	9 (1–137)	0.004
Death during ICU *n* (%)	49 (22.5)	9 (12.3)	40 (27.6)	0.010
Death during follow-up (total) *n* (%)	89 (40.8)	22 (30.1)	67 (46.2)	0.026
Mechanical ventilation *n* (%)	143 (65.6)	46 (63)	97 (66.9)	n.s.
Pre-existing diabetes *n* (%)	64 (29.4)	22 (30.1)	42 (29.0)	n.s.
BMI (m^2^/kg)	25.8 (14–86)	25.7 (15.9–40.5)	25.9 (14–86.5)	n.s.
Glucose (mg/dL)	137 (1–663)	150 (49–663)	133 (1–476)	n.s.
WBC (x10³/µL)	13.1 (0.1–208)	12.5 (1.8–29.6)	14 (0.1–208)	0.024
CRP (mg/dL)	100.5 (5–230)	17 (5–230)	164 (5–230)	<0.001
IL-6 (pg/mL)	150.0 (2–28000)	66.5 (1.5–5000)	250 (0.1–28000)	<0.001
Procalcitonin (ng/mL)	0.7 (0.03–207.5)	0.2 (0.03–100)	2.2 (0.1–207.5)	<0.001
Creatinine (mg/dL)	1.3 (0.1–15)	1.0 (0.2–15)	1.5 (0.1–10.7)	0.017
GFR-Cystatin C (mL/min)	34 (0–379)	59 (5–379)	21.5 (0–218)	<0.001
AST (U/L)	42 (5–20332)	47 (11–20332)	41 (5–7832)	n.s.
ALT (U/L)	30 (5–7867)	36.5 (7–7867)	25 (5–5890)	n.s.
γ-GT (U/L)	59 (5–1764)	56 (10–1764)	60 (5–5000)	n.s.
GLDH (U/L)	6 (1–5000)	8 (1–5000)	5 (1–686)	n.s.
AP (U/L)	82 (5–686)	77 (2–290)	86 (5–686)	n.s.
PCHE (U/L)	3997 (10–11001)	5189 (405–11001)	3698 (10–10896)	0.005
Bilirubin, total (mg/dL)	0.7 (0.1–20.8)	0.7 (0.1–20.8)	0.7 (0.1–18.9)	n.s.
Albumin (mg/dL)	28 (0.1–61.4)	29.1 (1.6–52.2)	27.1 (0.1–61.4)	n.s.
INR	1.16 (0.92–13)	1.17 (0.95–6.73)	1.16 (0.92–13)	n.s.
CTRP3 day 1 (ng/mL)	545.1 (82.9–2395.3)	758.8 (260.4–2269.1)	493.4 (82.9–2395.3)	<0.001

For quantitative variables, median and range (in parenthesis) are given. * Significance between sepsis and non-sepsis patients was assessed using the Mann–Whitney U-test (for quantitative variables) or chi-squared test (for categorical variables). APACHE: Acute Physiology And Chronic Health Evaluation; SOFA: sequential organ failure assessment; SAPS: simplified acute physiology score; BMI: body mass index; CRP: C-reactive protein; IL-6: interleukin 6; ICU: intensive care unit; INR: international normalized ration; WBC: white blood cell; GFR: glomerular filtration rate; AST: aspartate aminotransferase; ALT: alanine aminotransferase; γ-GT: gamma-glutamyltransferase; GLDH: glutamate dehydrogenase; AP: alkaline phosphatase; PCHE: cholinesterase; n.s.: not significant.

**Table 2 diagnostics-09-00063-t002:** Disease etiology of the study population leading to ICU admission.

	Sepsis	Non-Sepsis
	*n* = 145	*n* = 73
**Bacterial etiology of critical illness (sepsis)** Site of infection *n* (%)		
pulmonary	72 (50)	
abdominal	28 (19)	
urogenital	11 (8)	
other site of infection	34 (23)	
**Non-bacterial etiology of critical illness (non-sepsis)***n* (%)		
cardio-pulmonary disorder		29 (40)
acute pancreatitis		10 (14)
acute liver failure		4 (5.5)
decompensated liver cirrhosis		9 (12)
severe gastrointestinal hemorrhage		4 (5.5)
other non-bacterial etiology		17 (23)

**Table 3 diagnostics-09-00063-t003:** Correlations with CTRP3 plasma concentrations at ICU admission (Spearman rank correlation test).

	ICU Patients	
Parameters	*r*	** p*
Disease severity		
APACHE-II score	−0.063	0.378
SOFA score	−0.237	0.007
SAPS2 score	−0.187	0.129
Diabetes/insulin resistance		
Glucose	0.126	0.064
Glycosylated hemoglobin A1	0.051	0.671
C-peptide	−0.281	0.020
HOMA-β	−0.371	0.002
Inflammatory response		
White blood cell count	−0.148	0.029
Lymphocyte count	0.081	0.339
C-reactive protein	−0.390	<0.001
Procalcitonin	−0.207	0.009
TNF-α	−0.399	0.018
Interleukin-6	−0.266	0.001
Interleukin 10	−0.085	0.382
suPAR	−0.251	0.003
NTproCNP	−0.352	<0.001
Renal function		
Urea	−0.216	0.001
Cardiac function		
NTproBNP	−0.278	0.001
Liver function		
Protein	0.221	0.003
Albumin	0.330	<0.001
AST	0.102	0.147
ALT	0.048	0.483
Pseudocholinesterase	0.326	<0.001
Alkaline phosphatase	−0.145	0.039
aPTT	−0.156	0.023
Antithrombin III	0.240	0.006
D-dimers	−0.316	0.008
Lipid metabolism		
Cholesterol	0.213	0.005
LDL-cholesterol	0.326	0.008
HDL-cholesterol	0.387	0.001
Adipokines		
Resistin	−0.397	0.002
Leptin	0.258	0.055
RBP4	0.072	0.565
Adiponectin	−0.196	0.147
Ghrelin	−0.090	0.500
CTRP1	−0.075	0.270

SOFA: sequential organ failure assessment; SAPS2: simplified acute physiology score 2; HOMA-β: homeostasis model assessment-beta cell function; TNF-α: tumor necrosis factor-α; suPAR: soluble urokinase-type plasminogen activator receptor; NTproBNP: N-terminal pro C-type natriuretic peptide; NTproCNP: N-terminal pro B-type natriuretic peptide; CTRP1: C1q/tumor necrosis factor (TNF) related protein 1; RBP4: retinol binding protein 4; aPTT: activated prothrombin time; LDL: low-density lipoprotein; HDL: high-density lipoprotein; *—*p*-values less than 0.05 were considered as statistically significant.

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
