# Peer review of "Decreased CTRP3 Plasma Concentrations Are Associated with Sepsis and Predict Mortality in Critically Ill Patients"

_diagnostics, 2019, doi:10.3390/diagnostics9020063_

Round 1
Reviewer 1 Report
Diagnostics
Manuscript # 524711 (old J Clin Med 489399)
Title: Decreased CTRP3 plasma concentrations are associated with sepsis and predict mortality in critically ill patients
Review:
Overall:
In this manuscript the authors investigated CTRP3, an adipokine, as a novel biomarker of sepsis induced mortality. CTRP3 has been previously shown to play a role in metabolism, inflammatory diseases, liver function and in diabetes. Here the authors present data that suggest CTRP3 plasma concentrations are inversely correlated w/ inflammatory cytokines, as measure by plasma IL-6, and low CTRP3 levels are associated w/ mortality. While the authors have improved there submission, significant data clarification and presentation remains an issue for the reader to understand the authors conclusions.
Major Concerns:
A. While the authors state this is an ‘observational study’ w/ significant limitations, there remain concerns in the presentation of data making the authors conclusions hard to interpret.
Specific Comments:
1. Table 3. Please add an (*, symbol) to designate statistical significance. Is it possible that glucose (p = 0.064), leptin (p value 0.055) would correlate in a larger cohort ? These are both related to metabolism and are worth further discussion in the manuscript.
2. Table 2. This table remains confusing and needs to be broken out into 2 separate tables. 1 table should discuss the cause of sepsis bacterial (gram positive vs gram negative), fungal, viral etc. This would clarify the impact of LPS on the authors’ conclusions. The 2nd table should break out the co-morbidities like diabetes, smoking, heart disease, liver disease etc. Even if the findings are negative (diabetes, obesity) the results are informative to the reader.
3. Figure 1. The analysis of data in parts (b,c) remains confusing. Are the no septic no diabetic different to the septic no diabetic. Also are no septic diabetic different to the septic diabetic populations. The same for BMI in (c). There is no legend for (d) and is (d) just septic and if so please include non-septic to the graph.
4. Discussion- pg 10 line 215 ‘…compared to published cytokines such as CTRP1, ghrelin, etc’. These are not all ‘cytokines’ but soluble metabolic factors or soluble proteins.
5. Discussion: Inflammation and the complement/blood clotting systems go hand in hand. A discussion about CTRP3 and its correlation to these pathways (presented in Table 3) would enhance the manuscript.
Author Response
Response to Reviewer 1 Comments
Response to the reviewer:
Thank you very much for the thorough and fair review of our manuscript.
Reviewer #1:
Overall:
In this manuscript the authors investigated CTRP3, an adipokine, as a novel biomarker of sepsis induced mortality. CTRP3 has been previously shown to play a role in metabolism, inflammatory diseases, liver function and in diabetes. Here the authors present data that suggest CTRP3 plasma concentrations are inversely correlated w/ inflammatory cytokines, as measure by plasma IL-6, and low CTRP3 levels are associated w/ mortality. While the authors have improved there submission, significant data clarification and presentation remains an issue for the reader to understand the authors conclusions.
Major concerns:
A. While the authors state this is an ‘observational study’ w/ significant limitations, there remain concerns in the presentation of data making the authors conclusions hard to interpret.
Specific comments:
Table 3. Please add an (*, symbol) to designate statistical significance. Is it possible that glucose (p = 0.064), leptin (p value 0.055) would correlate in a larger cohort ? These are both related to metabolism and are worth further discussion in the manuscript.
Response:
We thank the reviewer for this important suggestion and in the Table 3 we have added an (*) symbol. P-values less than 0.05 were considered as statistically significant (see Table 3, pages 6-8, lines 160-161).
Low CTRP3 values may have diagnostic implications by pointing towards inflammatory and infectious diseases as well as prognostic implications by indicating a high risk of mortality. We could not find any associations to obesity and pre-existing diabetes as well as CTRP3 did not show any correlations with other key markers of glucose metabolism such as HbA1c, insulin or the homeostasis model assessment-insulin resistance (HOMA-IR). Thus, in ICU patients it would be potentially unlikely from this point of view that CTRP3 and glucose could be associated in larger cohorts. One can also assume for leptin that it seems to be in contrary to CTRP3 (not associated with sepsis but obesity and pre-existing diabetes (reference 11)) and therefore it is potentially not associated with CTRP3 when testing in larger cohorts. However, this important points warrants further investigation within larger patient cohorts and remains speculative from this point of view. As mentioned in the discussion section, the associations of CTRP3 are at this point descriptive and our findings need to be validated in larger, multi-center analysis (see page 12, lines 276-280).
Table 2. This table remains confusing and needs to be broken out into 2 separate tables. 1 table should discuss the cause of sepsis bacterial (gram positive vs gram negative), fungal, viral etc. This would clarify the impact of LPS on the authors’ conclusions. The 2nd table should break out the co-morbidities like diabetes, smoking, heart disease, liver disease etc. Even if the findings are negative (diabetes, obesity) the results are informative to the reader.
Response:
We thank the reviewer for this important suggestion. Unfortunately, we have only cumulative data on all microbes identified during the whole course of ICU treatment, which may not reflect the type of infection (gram+ vs. gram-) at the ICU admission, when the blood was sampled. Within the sepsis patients, the site of infection (e.g., pneumonia, bloodstream, abdominal, urogenital, others) did not affect CTRP3 concentrations, making it unlikely that the different type of bacteria had a major impact on CTRP3 levels. Thus, within the manuscript we described this aspect as
“Among the septic or non-septic critically ill patients, there was no association between CTRP3 plasma concentrations and these different disease aetiologies leading to ICU admission (data not shown).”
(see page 4-5, lines 131-133).
We presented in Table 2 only comorbidities that were considered as direct aetiology of critical illness. In line with this heart disease is given within the cardiopulmonary disorder subgroup. Diabetic patients are considered as ICU patients with metabolic alteration without being the reason for critical illness. Unfortunately, we do not have recorded smoking history of the ICU patients. Because of this, we have not identified smoking as a risk factor. It must be emphasized that it is difficult to document the smoking behaviour of ICU patients reliably when they are admitted.
With regard to liver diseases, we found only 9 patients with decompensated liver cirrhosis. Due to this fact, the number of patients with underlying exact liver disease was too low to allow a sufficient subgroup analysis.
Figure 1. The analysis of data in parts (b,c) remains confusing. Are the no septic no diabetic different to the septic no diabetic. Also are no septic diabetic different to the septic diabetic populations. The same for BMI in (c). There is no legend for (d) and is (d) just septic and if so please include non-septic to the graph.
Response:
We thank the reviewer for pointing out his/her suggestions regarding Figure 1. As the referee suggested, in Figures b and c we presented non-septic and septic ICU patients regarding to the presence of diabetes (obesity) classified by no/yes. In this context, all box-and-whiskers box plots represent different subgroups within the ICU-patients.
The Figure 4 legend is described in the Figure legend as: “ICU patients did not show significant differences in reference to BMI classification (d)”. In this figure we classified all ICU patients according to the presented BMI categories. However, we have now indicated in the Figure 4 the underlying ICU cohort as “all patients” (see Figure d, page 5).
Discussion- pg 10 line 215 ‘…compared to published cytokines such as CTRP1, ghrelin, etc’. These are not all ‘cytokines’ but soluble metabolic factors or soluble proteins.
Response:
We thank the reviewer for this important remark. We apologize for this error and have corrected this (see page 10, line 216).
Discussion: Inflammation and the complement/blood clotting systems go hand in hand. A discussion about CTRP3 and its correlation to these pathways (presented in Table 3) would enhance the manuscript.
Response:
We appreciate this comment and have provided additional discussion in the discussion section. As shown in Table 3, we found a broad variety of correlations with inflammation, coagulation and others. This is in line with the currently accepted characteristic of this member of the CTRP superfamily that has been documented to have a wide range of effects on metabolism, food intake, inflammation and vascular disorders. Thus, there are many aspects of CTRP3’s regulation and function that have to be explored (see page 12, lines 276-280).
Reviewer 2 Report
Authors investigated the compatibility of CTRP3 as the diagnostic as well as prognostic marker for sepsis among critically ill patients.
Fig. 1a is very impressive which showed significant decrease of CTRP3 in sepsis compared with non-sepsis critically ill patients as well as healthy controls. In addition, Fig.3c is also interest.
The following points should be answered by authors.
1) If the CTRP3 is a good candidate for diagnosis of sepsis, how about ROC curve and UC (area under curve) ? Then, is CTRP3 equivalent to other markers such as CRP, IL-6, NTproCNP which showed high correlation with CTRP3 ?
2) Similar to above-question, as the prognostic marker, how dominant in CTRP3 rather than CRP, IL-6, NTproCNP or other inflammatory markers which authors examined.
3) Regarding Fig. 3b, authors showed “upper 25%2, “25-50%” and “lower 25%”, where the “50-75” ?
4) Can authors compare CTRP3 plasma concentrations between sepsis-derived from pneumonia (pulmonary infection) (the most baseline disease in this study) and pneumonia without sepsis?
Author Response
Response to Reviewer 2 Comments
Response to the reviewer:
Thank you very much for the thorough and fair review of our manuscript.
Reviewer #2:
Authors investigated the compatibility of CTRP3 as the diagnostic as well as prognostic marker for sepsis among critically ill patients.
Fig. 1a is very impressive which showed significant decrease of CTRP3 in sepsis compared with non-sepsis critically ill patients as well as healthy controls. In addition, Fig.3c is also interest.
The following points should be answered by authors.
If the CTRP3 is a good candidate for diagnosis of sepsis, how about ROC curve and UC (area under curve) ? Then, is CTRP3 equivalent to other markers such as CRP, IL-6, NTproCNP which showed high correlation with CTRP3 ?
Similar to above-question, as the prognostic marker, how dominant in CTRP3 rather than CRP, IL-6, NTproCNP or other inflammatory markers which authors examined.
Response:
We sincerely thank this expert reviewer for his/her positive evaluation and the encouraging comments. We have conducted a direct comparison for CTRP3 and other markers such as CRP, IL-6 and NTproCNP by ROC curve analysis. CTRP3 had a partly equivalent and partly slightly lower AUC (0.709) for sepsis prediction compared to IL-6 (0.712), NTproCNP (0.704) and CRP (0.841). In particular, the ROC analyses of the biomarkers IL-6 (as the strongest indicator of endotoxin response) and NTproCNP are not significantly different from the AUC of CTRP3 (see attached Figure A in this response). These data have now been added to the revised manuscript.
Please see attached word document for Figure A. ROC analysis of CTRP3 compared with CRP, IL-6 and NTproCNP
Regarding Fig. 3b, authors showed “upper 25%2, “25-50%” and “lower 25%”, where the “50-75” ?
Response:
We thank the reviewer for her/his thoughtful question raised. CTRP3 concentrations were classified as lower (lowest 25%), middle (middle 25-50%), and upper (highest 25%) groups for the Kaplan-Meier survival curve analysis. Thus, high (highest 25%) vs. low (lowest 25%) CTRP3 plasma concentrations discriminated survival of the critically ill patients, as displayed by Kaplan-Meier survival curve analysis for CTRP3 separated into the mentioned quartiles.
Can authors compare CTRP3 plasma concentrations between sepsis-derived from pneumonia (pulmonary infection) (the most baseline disease in this study) and pneumonia without sepsis?
Response:
We would like to thank the reviewer for raising this important point. Unfortunately, we have only cumulative data on all microbes identified during the whole course of ICU treatment, which may not reflect the type of infection (gram+ vs. gram-) at the ICU admission, when the blood was sampled. Within the sepsis patients, the site of infection (e.g., pneumonia, bloodstream, abdominal, urogenital, others) did not affect CTRP3 concentrations, making it unlikely that the different type of bacteria had a major impact on CTRP3 levels. Thus, within the manuscript we described this aspect as
“Among the septic or non-septic critically ill patients, there was no association between CTRP3 plasma concentrations and these different disease aetiologies leading to ICU admission (data not shown).”
(see pages 4-5, lines 131-133).

This manuscript is a resubmission of an earlier submission. The following is a list of the peer review reports and author responses from that submission.
Round 1
Reviewer 1 Report
Dear Authors
Authors suggested CTRP3 as a new biomarker for the diagnosis of sepsis and prediction of mortality. Adiopokine family is an interesting topic in critical care. However, the clinical implication of CTRP3 is weak because results were not significantly different with previous data about other adipokine family. I think it would increase the value of manuscript if authors will show the data about the strengths of CTRP3 than other adipokine families.
It is difficult to explain the association with the initial CTRP3 plasma level and long-term survivor. To support the long-term outcome, the changes of CTRP3 level or the level at discharge are needed.
Reviewer 2 Report
Manuscript # 489399
Title: Decreased CTRP3 plasma concentrations are associated with sepsis and predict mortality in critically ill patients
Review:
Overall:
In this manuscript the authors investigated CTRP3, an adipokine, as a novel biomarker of sepsis induced mortality. CTRP3 has been previously shown to play a role in metabolism, inflammatory diseases, liver function and in diabetes. Here the authors present data that suggest CTRP3 plasma concentrations are inversely correlated w/ inflammatory cytokines, as measure by plasma IL-6, and low CTRP3 levels are associated w/ mortality. While the data is novel, the data is also observational with results limited to general statistical correlations in a post hoc type of analysis.
Major Concerns:
A. CTRP3 has been shown to be a metabolic marker by multiple groups (Peterson JM. 2010. JBC). An increase in CTRP3 results in a decrease in glucose. There is literature suggesting that ICU patients especially septic patients have poor nutrition. New data also indicates that activated T cells use glucose as an energy source, was 1) glucose or other nutritional levels measured and 2) T cell function tested ? One might predict that low CTRP3 levels that associate w/ mortality is a measure of overall nutritional status.
B. Comparing and contrasting CTRP3 over time and to additional adipokines like leptin, RBP4 as well as resistin would help to clarify the specificity of CTRP3 to sepsis and specifically morbidity and mortality.
Specific Comments:
1. Patient population. Were co-morbidities defined and statistically analyzed ? In addition, were co-morbidities or previous history of liver disease, cirrhosis, or diabetes obtained and statistically tested.
2. Methods: In addition to the plasma measure of CTRP3, the day clinical blood tests were taken and analyzed needs to be clearly stated.
3. Table 1. Include race as part of the demographics. Include the SOFA score along w/ the APACHE-II score. Glucose values or another measure of nutritional state, liver function tests individually, and co-morbidities (diabetes, heart disease, smoking/COPD, alcoholism, liver disease etc) should also be presented. Statistical tests should be performed w/ P values for tests between groups presented on the right side of the table. Healthy controls should be added as part of the table (age/gender/race/BMI etc).
4. Table 2. Aetiology of Sepsis critical illness. This is confusing the way it is presented. Break out bacterial (types), vs non-bacterial, vs unknown etiology. In both the introduction and discussion the impact of LPS on CTRP3 is presented. Therefore the addition of LPS related data based on each patient population would significantly enhance the authors conclusions.
5. Figure 1. As the ‘N’ is fairly large in the groups, would presenting a single graph using healthy controls, critically ill non-septic and septic more clearly illustrate the data. Is there a difference in glucose levels or liver function tests and CTRP3 levels. Please state in the methods why a BMI of 30 was used and do these results hold true if you use additional BMI cut-offs like<18 (underweight), 18-25 (healthy), 25-29 (overweight). For (c,d) could the data be broken out by septic and non-septic as this might more reflect changes specifically in sepsis. Also was CTRP3 tested over time of the ICU stay as we know that metabolic, inflammatory and immune system changes occur over the course of sepsis.
6. Table 3. As CTRP3 is associated w/ metabolism, adding Glucose and additional liver function tests (AST, ALT) would enhance the interpretation of the data. Please add to the methods when these clinical tests were assessed (at ICU admission, over time and discharge etc). One could also perform multiple associations at various times of the ICU stay using a more complex statistical model.
7. Figure 2. Please indicate the type of statistical test in the figure legend and the day of blood draw. The IL-10 data should be added as well as the addition of TNF-a levels.
8. Figure 3. Please discuss why 620.6 ng/mL is an “optimal cut-off”. What value or range is ‘normal’ vs an indicator of other metabolic disorders (diabetes, liver disease etc). A similar analysis using an additional adipokine like resistin would clarify the specificity of CTRP3 in sepsis. Also was the data broken out by septic and non-septic as this might more reflect changes specifically in sepsis.